



# Remote sensing of methane plumes: instrument tradeoff analysis for detecting and quantifying local sources at global scale

Siraput Jongaramrungruang[1], Georgios Matheou[2], Andrew K. Thorpe[3], Zhao-Cheng Zeng[1], and Christian Frankenberg[1,2]

[1]Division of Geological and Planetary Sciences, California Institute of Technology, Pasadena, CA 91125, USA
[2]Department of Mechanical Engineering, University of Connecticut, Storrs, CT 06269, USA
[3]NASA Jet Propulsion Laboratory, California Institute of Technology, Pasadena, CA 91109, USA

**Correspondence:** Siraput Jongaramrungruang (siraput@caltech.edu); Christian Frankenberg (cfranken@caltech.edu)

**Abstract.** Methane ($CH_4$) is the 2nd most important anthropogenic greenhouse gas with a significant impact on radiative forcing, tropospheric air quality and stratospheric water vapor. Remote-sensing observations enable the detection and quantification of local methane emissions across large geographical areas, which is a critical step for understanding local flux distributions and subsequently prioritizing mitigation strategies. Obtaining methane column concentration measurements with low noise and

minimal surface interference has direct consequences for accurately determining the location and emission rates of methane sources. The quality of retrieved column enhancements depends on the choices of instrument and retrieval parameters. Here, we studied the changes in precision error and bias as a result of different spectral resolutions, instrument optical performance and detector exposure times by using a realistic instrument noise model. In addition, we formally analysed the impact of spectrally complex surface albedo features on retrievals using the Iterative Maximum a Posteriori- Differential Optical Absorption

Spectroscopy (IMAP-DOAS) algorithm. We built an end-to-end modelling framework that can simulate observed radiances from reflected solar irradiance through a simulated $CH_4$ plume over several natural and man-made surfaces. Our analysis shows that complex surface features can alias into retrieved methane abundances, explaining the existence of retrieval biases in current airborne methane observations. The impact can be mitigated with higher spectral resolution and a larger polynomial degree to approximate surface albedo variations. Using a spectral resolution of 1.5 nm, an exposure time of 20 ms, and a

polynomial degree of 25, a retrieval precision error below $0.007 \, \mathrm{mole \, m^{-2}}$ or 1.0 % of total atmospheric $CH_4$ column can be achieved for high albedo cases, while minimizing the bias due to surface interference such that the noise is uncorrelated among various surfaces. At coarser spectral resolutions, it becomes increasingly harder to separate complex surface albedo features from atmospheric absorption features. Our modelling framework provides the basis for assessing trade-offs for future remote-sensing instruments and algorithmic designs. For instance, we find that improving the spectral resolution beyond 0.2 nm would

actually decrease the retrieval precision as detector readout noise will play an increasing role. Our work contributes towards building an enhanced monitoring system that can measure $CH_4$ concentration fields to determine methane sources accurately and efficiently at scale.





## 1 Introduction

Anthropogenic greenhouse gas emissions have been rising continuously, affecting the global climate and the environment (Stocker et al. (2013)). Among the most important anthropogenic emissions are carbon dioxide ($CO_2$) and methane ($CH_4$). Due to a much shorter lifetime of $CH_4$ ($\approx 9$ years) compared to $CO_2$ ($\approx 500$ years), $CH_4$ has gained attention as target for mitigation efforts to achieve short- and medium-term reductions of global warming (Montzka et al., 2011; Prather et al., 2012; Shindell et al., 2012). In general, anthropogenic methane emissions are also much more uncertain than those of carbon dioxide, which can often be characterized to within approximately 10% just from budget assumptions (Gurney et al., 2019). For instance, just the question whether or not the leak rate in the natural gas extraction system is 1 or 2% is equivalent to a 100% uncertainty in methane emissions. At the same time, leak rate outliers (Frankenberg et al., 2016; Duren et al., 2019; Cusworth et al., 2021) are often local in nature and easily fixable, representing a win-win scenario if faulty equipment or practices can be readily detected and then efficiently mitigated. As $CH_4$ reduction plays a significant role in climate mitigation efforts, one key step in emission reduction is determining where these emissions are coming from. This is underpinned in the 2018 NASA Decadal Survey, which calls out the identification and understanding of $CH_4$ emissions as one of the top priorities in the efforts to improve future climate projections, and help lead the way in emission reduction (National Academies of Sciences and Medicine, 2018).

Remote-sensing instruments using absorption spectroscopy have emerged as one promising solution for measuring atmospheric $CH_4$ concentration over large geographical areas. Space-based $CH_4$ retrieval techniques from satellite observations such as the SCanning Imaging Absorption SpectroMeter for Atmospheric CHartographY (SCIAMACHY, (Frankenberg et al., 2005a, 2011)) and the Greenhouse gases Observing SATellite (GOSAT, (Parker et al., 2011, 2015; Turner et al., 2015)) were dedicated missions with $CH_4$ as a key target. They used $CH_4$ absorption features in the 1.6 and 2.3 μm bands to retrieve column $CH_4$ concentration across the globe. The TROPOspheric Monitoring Instrument (TROPOMI) with a spatial resolution of a few kilometers has also been shown to be capable of identifying regions of high emissions (de Gouw et al., 2020; Hu et al., 2018). These satellites, which have been designed by the atmospheric community, have particular sets of goals and instrument specifications that are mostly targeted towards obtaining regional-scale methane distributions with high accuracy and precision. Most of these satellites were designed to measure gradients of methane concentration across hundreds to thousands of kilometers of scale, as this enables surface flux inversions at the global scale. Typically, all of these instruments have one feature in common — they have very high spectral resolution (0.05-0.25 nm) to distinguish individual methane absorption lines from spectrally smooth surface albedo variations. However, due to their coarse spatial resolutions, the measurements are not yet at a level where local sources can be identified, attributed to a specific source type (e.g. compressor station or well pad) and mitigated directly.

One potential solution to fill this scale gap is using an airborne instrument that has much higher spatial resolution such as the Methane Airborne MAPper (MAMAP, Gerilowski et al. (2011)) or the next-generation Airborne Visible/Infrared Imaging Spectrometer (AVIRIS-NG, Thorpe et al. (2017)). The latter is based on the insight that methane column enhancements at high spatial resolution (a few meters) can be so high that the retrieval of the absorbing feature can be done even with moderate spectral resolution (5-10 nm). If the methane column is expressed similar to the Dobson unit, i.e. as the thickness of a layer of



pure gas which would be formed by the total column amount at standard conditions, the layer thickness at current background methane conditions would only be about 1.6 cm. Thus, a pure methane layer of only 1.6 mm would enhance the total column by 10%, which is certainly realistic for measurements of methane point sources at fine spatial resolution.

Bradley et al. (2011) and Thorpe et al. (2014) were among the first to show that moderate resolution instruments can detect methane plumes, even when the strong 2.3 μm methane band is convolved with the AVIRIS (10 nm) or AVIRIS-NG (5 nm) instrument line-shape functions. While individual lines are hard to resolve, the strong methane band in this range causes enough fine-structure in terms of bulk absorptions by a multitude of methane lines within the instrument resolution. Previous studies by Frankenberg et al. (2016), Duren et al. (2019) and Cusworth et al. (2021) have utilized AVIRIS-NG to conduct field campaigns in California and the Four-Corner region where they could create a map of methane enhancements in the area and detected several hundreds of individual methane sources, which followed a heavy-tail flux distribution. The concept of this airborne spectrometer provides a signal of opportunity for local source detection and quantification. However, the instrument was not originally designed for methane detection and it does not meet the same precision and accuracy requirements as those satellites from the atmospheric community for methane retrieval at a global scale, which require accuracy better than 1%, equivalent to enhancements of about 19 ppb in $XCH_4$ (or $4 \cdot 10^{17}$ molec cm$^{-2}$, $0.007$ mole m$^{-2}$ or $152$ ppm − m). One significant drawback of coarse spectral resolution is the occurrence of retrieval artefacts that often correlate with specific surface features (see Figure 1). This can confound the detection and quantification of methane point sources in the analysis and obviate the robust detection of subtle gradients at larger spatial scales (Jongaramrungruang et al. 2021 (in review)). Even though most strong plumes can be observed, the uncertainties in the overall detection and quantification at the regional level can present persistent problems and often involve human judgement to isolate plumes from artefacts. In fact, during each of the California survey, the Four-corner study, and the Permian survey (Duren et al. (2019), Frankenberg et al. (2016), Cusworth et al. (2021)), human analysts were involved in a manual process to look through each flight line to classify true emission sources from false positives. Similarly, in previous space-based studies to locate and approximate a large emission such as a blowout event, prior information about the location of the source is usually already known, making it much easier to find a true methane source from space-based measurements over the area after the fact. There are ongoing efforts to develop automated plume detection for existing instruments. Future real-time monitoring systems would greatly benefit from next generation instruments that would reduce retrieval artifacts and provide retrievals with improved accuracy, such that remote-sensing measurements can be analyzed to locate and quantify plumes automatically, at scale. Hence the origination of this study.

If we had the opportunity to design a new instrument that is optimized for methane retrievals at fine spatial resolution (sub 50-m), what would the specifications of this instrument look like? Thorpe et al. (2016) proposed a 1 nm instrument to mitigate the drawbacks of AVIRIS-NG. To fully evaluate optimal performance metrics, we have to consider the tradeoff between spectral and spatial resolutions and concomitant changes in detector noise characteristics. On the one hand, the instrument needs to meet the requirements of the atmospheric community so that it can unambiguously differentiate methane from other confounding factors. On the other hand, the instrument should have adequate integration time to achieve high spatial resolution with sufficient signal to noise levels. Here, we investigate this tradeoff and evaluate risks and benefits for methane retrievals at fine resolutions with the purpose of successfully detecting and quantifying local sources in mind. We built an end-to-end



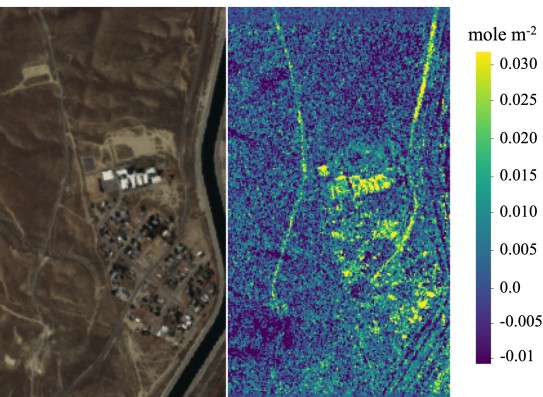

**Figure 1.** Example of systematic outliers from a retrieved AVIRIS-NG scene (right) compared with an RGB image (left).

modeling framework that can generate reflected solar radiance through a methane plume of known concentration over realistic surfaces, and perform the retrieval from the corresponding observed radiance under a given instrument to output the predicted methane concentration in each column. Our model calculates the noise-equivalent spectral radiance (NESR) as a function of incoming radiance and instrument parameters such as integration time, detector size, quantum efficiency, readout noise, and spectral resolution, rather than prescribing the signal-to-noise ratio (SNR) as an independent variable. By varying the instrument and retrieval parameters, we can derive the associated precision error and bias from the retrieval. We also compare the tradeoff between the two most frequently used fitting windows in the 1.6 and 2.3 μm ranges.

Section 2 outlines the background on radiative transfer, followed by data and methodology on the forward model with realistic surface reflectances, instrument operators and retrieval setups. Results and discussion are provided in Section 3. The final section contains concluding remarks and future steps.

## 2 Data and Methodology

For the sake of simplicity, we ignore the impact of atmospheric scattering, as Rayleigh scattering is negligible in the near-infrared and the impact of aerosols is rather small compared to methane enhancements in the near-field of local sources. While aerosols can cause small systematic biases in the retrieved methane amount, their impact on measuring anomalies caused by methane plumes should be rather small. In addition, the precision error is not strongly affected by neglecting atmospheric scattering and experience with previous moderate resolution methane mapping has shown that surface interferences are more crucial. In the absence of atmospheric scattering and assuming a Lambertian surface, the reflected radiance as measured by an instrument at the top of the atmosphere in the Nadir direction can be modelled as

$$L_\lambda = I_{0,\lambda} \cdot r_\lambda \cdot T_{\lambda\uparrow} \cdot T_{\lambda\downarrow} \cdot \frac{\cos(\text{SZA})}{\pi} \tag{1}$$





where $I_0$ stands for the incoming solar irradiance spectrum, $T_{\lambda\downarrow}$ the atmospheric transmission along the photon light-path downwards to the surface, $r_\lambda$ the surface albedo, $T_{\lambda\uparrow}$ the transmission along the light-path on the way up from the surface to the instrument, and $SZA$ the solar zenith angle. Figure 2 illustrates a schematic for equation 1. The subscript $\lambda$ denotes the

wavelength dependence of these variables. The multiplication in equation 1 is element-wise for each wavelength in the spectral range of interest.

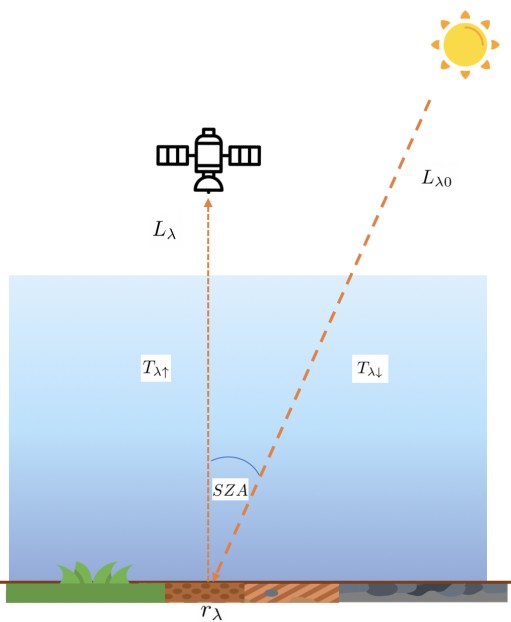

**Figure 2.** A schematic for reflected sunlight from the sun through the atmosphere to a spectrometer in space.

## 2.1   Incoming solar irradiance

We constructed $I_{0,\lambda}$ by multiplying a continuum level spectrum with a high-resolution solar transmission spectrum that includes absorption features in the sun's photosphere, so-called Fraunhofer lines. These absorption features are caused by trace elements

in the solar photosphere. The continuum spectrum is obtained from Meftah, M. et al. (2018) with 0.2 nm resolution. We fitted a $3^{rd}$-order polynomial to this measured spectra in a 1.4 - 2.5 µm range to obtain a smooth continuum spectrum. A disk integrated solar transmission spectrum is obtained from a tabulated line-list compiled by Toon (2015). We interpolated the baseline and transmission spectra to a common 0.01 nm resolution grid and multiplied them to obtain a high-resolution solar irradiance $I_{0,\lambda}$.



## 2.2 Atmospheric transmission

The atmospheric transmission can be modelled using the Lambert-Beer Law by dividing the atmosphere into vertical layers, each with constant pressure, temperature and gas number density. We calculate absorption cross-sections for each layer using the HITRAN spectral database (Gordon et al., 2017) and a Voigt lineshape. The transmission then reads

$$T_\lambda = \exp\left( - \left( \sum_i \sum_k n_{i,k}\, \sigma_{i,k(\lambda)} \times AMF_k \right) \right), \tag{2}$$

where $i$ denotes the $i^{th}$ gas species, $k$ the $k^{th}$ layer, $n$ the vertical column density (molecules cm$^{-2}$), and $\sigma$ the gas absorption cross-section (which is a function of pressure ($P$) and temperature ($T$)). The air mass factor ($AMF$) per layer denotes the ratio of the integrated number concentration along the actual photon light-path and the geometric vertical integration. In the absence of scattering, it is $1/cos(SZA)$ for the incoming light at SZA, and equal to 1 for the outgoing light as seen in Nadir.

The transmission of the atmosphere is calculated from the background gas concentrations from the top of the atmosphere (TOA) to the surface. In addition, we consider the enhancements due to local gas emissions which we primarily considered to reside between the atmospheric boundary layer (BL) and the surface. The instrument is assumed to be located at the TOA.

For the background transmission, we divided the atmosphere into 72 layers, and used an atmospheric profile for $p$ and $T$ from the Four-Corners area (lat = 36.8°, lon = -108°). We considered H$_2$O, CO$_2$ and CH$_4$ in the background. Concentration of H$_2$O is obtained from the MERRA reanalysis (Rienecker et al., 2011) vertical profile. For simplicity, background CO$_2$ and CH$_4$ are set to volume mixing ratios of 400 ppm and 2000 ppb, respectively. For the gas enhancement within the BL, 300 vertical layers are used to model a simulated 3D methane plume enhancement from Large Eddy Simulation (LES) output. The LES enables a realistic simulation of how methane concentrations from a point source evolve in space and time, as it generates the time-resolved three-dimensional CH$_4$ distribution in the boundary layer. The full description of the LES model setup for CH$_4$ plume emanating from a point source can be found in Matheou and Bowman (2016), with the model parameterization and initialization detailed in Jongaramrungruang et al. (2019). We computed the gas absorption cross sections in each layer using an open-source Julia radiative transfer tool that calculates the cross-section efficiently using the HITRAN database and GPU capability (Gordon et al., 2017). A Voigt absorption line shape is used in our study. We note that we used 300 vertical layers in the BL in the full forward model to simulate the observed outgoing spectra, but will use a much smaller number of layers in the retrieval step (more details in section 2.4).

### 2.2.1 Surface Reflectance

To analyze the impact of surface spectral features, we compiled a database of different surface albedos from the ECOSTRESS spectral library (Meerdink et al., 2019) to investigate the impact on our traditional retrieval technique that we use from space, in which the surface is typically characterized by a low-order polynomial in wavelength. In fact, many spectroscopic measurement techniques rely on the fact that atmospheric features exhibit sharp absorption features while surface are spectrally smooth (Platt and Stutz, 2008). From a physical perspective, this is related to more rapid quenching of an excited state in solids or liquids as well as the suppression of rotational energy levels. At the same time, this separation of high-frequency atmospheric features





from low-frequency surface features is at the core of our study, as instruments such as AVIRIS(-NG) have spectral resolutions that can blur the separation between frequencies, allowing surface features to alias into methane retrievals.

The compiled database contains more than 2000 surfaces over 5 main categories of rock, soil, mineral, photosynthetic and non-photosynthetic vegetation, and man-made construction materials. Examples of these surface albedos near the 2.3 µm $CH_4$ absorption range are shown in Figure 3. The typical spectral resolution of the database is 2 nm and we resampled all spectra to a common grid, subsequently used with spline interpolation in our high-resolution forward model at 0.01 nm.

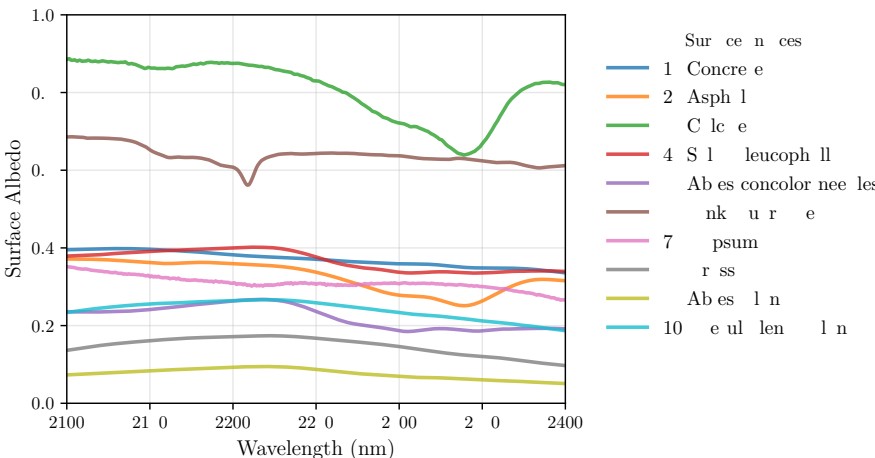

**Figure 3.** Albedo spectral variations near 2.3 µm from distinct surface reflectances in our database.

Based on this database, we can create an arbitrarily diverse set of surfaces underlying the simulated 3D methane field. For example, a checkerboard-styled tile consisting of $3 \times 10$ surfaces can be constructed before we overlay the 3D plume. A retrieval can then be applied for each pixel across the source to visualize the impact on precision error and bias caused by different surfaces. We also used LANDSAT data (Wulder et al., 2019) to represent a natural distribution of surfaces in the Durango area, Colorado. Although it is a 30-m resolution, we can up sample the surface grids to 5-m each. At each pixel, we matched the measured surface albedo from LANDSAT observations at 0.48, 0.56, 0.65, 0.87, 1.61, and 2.20 µm to the closest possible surface in our database, and queried its full albedo spectra for our simulations.

## 2.3 Instrument operators

### 2.3.1 Convolution and observed spectra

The actual observed spectrum that is recorded by the instrument is the convolution of the high-resolution incident light $L_\lambda$ with the instrument line shape, denoted here as instrument kernel. This convolution is performed in the intensity space given by

$$< L(\lambda) >= \int_{-\infty}^{\infty} L(\lambda') \, \phi(\lambda - \lambda') \, d\lambda' \tag{3}$$





where $L(\lambda)$ is the incident spectra on the device and $<>$ denotes the convolution with the instrument kernel $\phi$. The convolved spectrum can then be interpolated and resampled to the output wavelength grids ($\lambda_{out}$) of the instrument, in an observing spectral range of interest. This output spectra is used as the measurement vector in the retrieval process (more details in section 2.4).

The instrument kernel $\phi$ is modelled using a Gaussian distribution with zero mean and a given Full-Width-Half-Maximum

(FWHM). In our experiment, we treated the FWHM as an independent variable that varies between 0.04 and 10.0 nm. The FWHM is a key property of an instrument that determines what spectral variations can be resolved. For instance, if the spectral resolution is coarser than the rotational fine-structure of a vibrational-rotational absorption band, the P and R branches of this band will appear as just two separate broad-band absorption features. The spectral sampling interval (SSI) varied accordingly with FWHM. Here, we use two cases of SSI equal to FWHM/2.5 (near Nyquist sampling in atmospheric sounders) and

FWHM/1.0 (critical sampling as in AVIRIS-type imaging spectrometers).

### 2.3.2 Noise-equivalent spectral radiance

Towards designing the optimal instrument, we have to evaluate the trade space of spectral resolution, spatial resolution and detector characteristics. Previous studies evaluated the trade space between signal to noise ratio (SNR) and spectral resolution, treating the SNR as an independent variable when varying the spectral resolution (Thorpe et al., 2016; Cusworth et al., 2019;

Ayasse et al., 2019). However, the SNR deteriorates at higher spectral resolution, as fewer photons are being counted by each detector pixel. To evaluate this properly, we start working from an instrument model directly where the noise is a function of incoming radiance, and parameters such as integration time, F number, detector pixel and then the spectral resolution as described in Strandgren et al. (2020). In this approach, the SNR will be a dependent variable based on our instrument specifications and the actual observed radiance.

The electronic signal measured by each detector pixel can be expressed as (Strandgren et al., 2020):

$$S = <L_\lambda> \frac{\pi A_{det}}{4 f_{num}^2} \cdot \eta \cdot Q_e \cdot \Delta\lambda \cdot t_{int}, \tag{4}$$

where $<L_\lambda>$ is our simulated radiance, $A_{det}$ the detector pixel area, $f_{num}$ the instrument's f number, $\eta$ the optical efficiency of the spectrometer, $Q_e$ the quantum efficiency of the detector, $\Delta\lambda$ the SSI and $t_{int}$ the integration time.

For the NESR, we consider two dominant noise terms, namely shot noise (proportional to $\sqrt{S}$) and effective readout noise

$\sigma_{ro}$:

$$NESR = \sigma_{L_\lambda} = \sqrt{S + \sigma_{ro}^2} \tag{5}$$

For the analyses in this study, we varied FWHM from 0.04 nm to 10.0 nm, integration time from 5 ms to 105 ms, with other default parameter setups as described in Table 1, unless stated specifically otherwise.





**Table 1.** A table for default parameter settings in our simulations.

| Parameter | Value |
|-----------|-------|
| Integration time | 20 ms |
| Detector size | 30.0 $\mu$m |
| F-number | 2.4 |
| Quantum efficiency | 0.95 |
| Optical bench efficiency | 0.5 |
| Readout noise | 100.0 |

## 2.4 Retrieval setup

### 2.4.1 Forward model

The retrieval forward model is similar to that described earlier in previous sections. The main difference is that we now treat the methane concentration in the boundary layer as elements of the state vector that we want to retrieve, as the methane profile is unknown during the observation. A much smaller number of layers to represent methane enhancements within the boundary layers is used. This is an important consideration when dealing with moderate spectral resolution since the information content is not high enough to discriminate between different layers. Polynomial terms are also used to represent the change in surface reflectance with wavelength. The forward model can be written mathematically as

$$\boldsymbol{y}_\lambda = F(\boldsymbol{x}) = <L_{\lambda_0} \cdot T_{\lambda\uparrow} \cdot T_{\lambda\downarrow}> \underbrace{\sum_{d=1}^{D} a_d P^d(\lambda)}_{\approx r_\lambda} \tag{6}$$

where $\boldsymbol{y}$ is the measurement vector, $P^d(\lambda)$ a Legendre polynomial term at degree $d$, $a_d$ a coefficient for each $P^d(\lambda)$, $D$ the number of polynomial degree used in the retrieval. To evaluate the polynomial degree within the fitting range, we converted the wavelength range within the fitting window to span -1 through 1. Here, the state vector $\boldsymbol{x}$ consists of the vertical column density (molecules cm$^{-2}$) of the respective gases in different layers, and polynomial coefficients accounting for low-frequency surface features.

In our experiment, we set the number of water vapour and methane concentration layers to be 2 and 10 respectively. We vary the number of polynomial degree, $D$, to investigate its ability to disentangle the surface spectral variations from the atmospheric features, while simultaneously considering different instrument spectral resolutions. Spectrally smooth surface albedos might only require a few polynomial coefficients while more complex surface features within the 260 nm fitting window can require more than 25 coefficients, as will be shown later.





### 2.4.2 Optimal estimation

To solve for the optimal state vector from this nonlinear system, we used the Iterative Maximum a Posteriori- Differential

Optical Absorption Spectroscopy (IMAP-DOAS) approach (Frankenberg et al., 2005a). Based on maximizing the a posteriori probability density function as introduced to the atmospheric community by Rodgers (2000), the iterative solution can be written as

$$\boldsymbol{x}_{i+1} = \boldsymbol{x}_a + (\mathbf{K_i}^T \mathbf{S}_\epsilon{}^{-1} \mathbf{K_i} + \mathbf{S_a}^{-1})^{-1} \mathbf{K_i}^T \mathbf{S}_\epsilon{}^{-1} \cdot [\boldsymbol{y} - F(\boldsymbol{x}_i) + \mathbf{K_i}(\boldsymbol{x}_i - \boldsymbol{x}_a)] \tag{7}$$

where $\boldsymbol{x}_i$ is the state vector at the $i^{th}$ iteration, $\boldsymbol{x}_a$ a priori state vector, $\mathbf{S}_\epsilon$ the measurement error covariance matrix, $\mathbf{S_a}$ a priori

covariance matrix, $\mathbf{K_i}$ the Jacobian of the forward model evaluated at $\boldsymbol{x}_i$. $F(\boldsymbol{x}_i)$ stands for a forward model at each $\boldsymbol{x}_i$. When consecutive changes in the reduced $\chi^2$ of the fit drop below a tolerance level of $10^{-3}$, we stop iterations. The Jacobian matrix $\mathbf{K_i}$ is computed analytically in each $i^{th}$ iteration using automatic differentiation techniques. The a-priori covariance matrix $\mathbf{S_a}$ helps constrain the fit based on the possible range of concentration (thus $\boldsymbol{S}_a$ is a square matrix with a size equal to the length of $\boldsymbol{x}_i$). Here, we use loose prior constraints, thus having no significant impact on the retrieved total columns or posterior

errors. The measurement error covariance matrix $\mathbf{S}_\epsilon$ is a matrix, in which the diagonal elements are the estimated variances of instrument noise at the observed wavelength grid $\lambda_{out}$ (thus $\mathbf{S}_\epsilon$ is a square matrix with a size equal to the length of $\lambda_{out}$). These variances are computed from the instrument noise model outlined in Section 2.3.2.

### 2.4.3 Error estimations

The posteriori error estimate $\hat{\mathbf{S}} = (\mathbf{K}^T \mathbf{S}_\epsilon \mathbf{K} + \mathbf{S_a}^{-1})^{-1}$ provides the full error covariance matrix of the retrieved state vector

$\hat{\boldsymbol{x}}$. The quantity of interest in our application for $CH_4$ detection and quantification is the total column concentration of $CH_4$. To obtain this quantity from our retrieval, we can find the summation of the state vector elements over indices corresponding to $CH_4$. We can thus define a summation operator $\boldsymbol{h}$ of the same size as the state vector, filled with ones where $CH_4$ state vector elements are located and zeros elsewhere. The summation of the total column is readily derived as $\boldsymbol{h}^T \hat{\boldsymbol{x}}$ and the variance of the total column is computed as $\boldsymbol{h}^T \hat{\mathbf{S}} \boldsymbol{h}$ (Rodgers, 2000). The bias error is obtained as the difference between the best estimated

value in the absence of instrument noise and the true $CH_4$ vertical column enhancement.

### 2.5 LES $CH_4$ plumes

The LES is used to generate the time-resolved three-dimensional $CH_4$ distribution in the boundary layer. This provides a realistic distribution of how methane concentrations from a point source on the ground evolve across the area. The flow in the boundary layer is driven by a constant geostrophic wind in the x direction, influencing the shape of the plume. We conducted

LES experiments with a geostrophic wind speed of $4 \mathrm{\ m\ s^{-1}}$ with a range of emission rates from 50 to $5000 \mathrm{\ kg\ h^{-1}}$.

To simulate the reflected sunlight over the area of a $CH_4$ emission source, we use the output from LES that has a realistic 3D concentration field of $CH_4$ at a prescribed emission rate and overlay this on top of surface tiles of different albedos. Within each tile, the albedo is chosen from a distinct surface in our database. Figure 4 illustrates this conceptual setup.




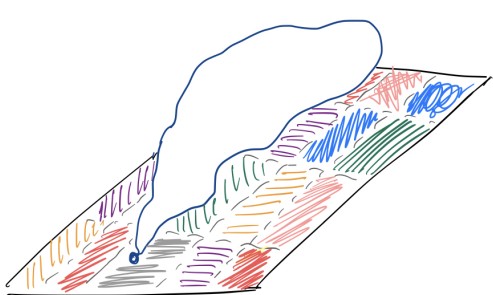

**Figure 4.** A schematic showing the setup for a methane plume over different surface tiles, each of which contains one surface albedo from our database.

At each pixel, the high-resolution outgoing reflected radiance can be calculated and subsequently be convolved with the instrument kernel. This yields the simulated observed radiance at each pixel that we implemented as a measurement vector. Accordingly, the pixel radiances are then converted to signal strength in electrons and respective noise levels using our instrument model. The IMAP-DOAS algorithm is applied to retrieve the column $CH_4$ and provide error estimates. We can vary the instrument parameters as well as the number of polynomial degrees in the retrieval to explore their relationship with the associated errors.

## 3   Results and Discussion

### 3.1   Simulated high-resolution and observed radiance

As a first step towards understanding the effect of the instrument spectral resolution, we simulated high-resolution spectrum $< L_\lambda >$ in the 1.6 μm and 2.3 μm bands and their corresponding Jacobians for instruments with FWHM of 0.2, 1.5, 5.0 and 10.0 nm. This simulation is based on $CH_4$ concentration profile near an origin of the methane plume with an emission rate of 200 kg hr$^{-1}$, over a construction concrete (albedo index 1 in Figure 3), with simulations shown in Figure 5. As the instrument FWHM increases, individual absorption lines are increasingly blurred and the less high-resolution absorption features are recorded. The Jacobian represents the change in radiance with respect to the changes in gas concentration. Here we show the Jacobian for $CH_4$ and $H_2O$ close to the ground. For the 1.6 μm band, the total radiance is in the range of 20-30 mW m$^{-2}$ nm$^{-1}$ sr$^{-1}$, and we observed the strongest absorption feature between 1.66-1.67 μm, covering the Q-branch in the $2\nu^3$ band. On the other hand, for the 2.3 μm band, the radiance varies from around 10 to 3 mW m$^{-2}$ nm$^{-1}$ sr$^{-1}$ across the band. The $CH_4$ absorption features are much stronger and prominent over a wider range from 2.2 to 2.4 μm. To compare the two fitting windows in terms of their effectiveness in $CH_4$ measurements, we explore errors associated with the $CH_4$ retrieval using each of these bands in the next section.





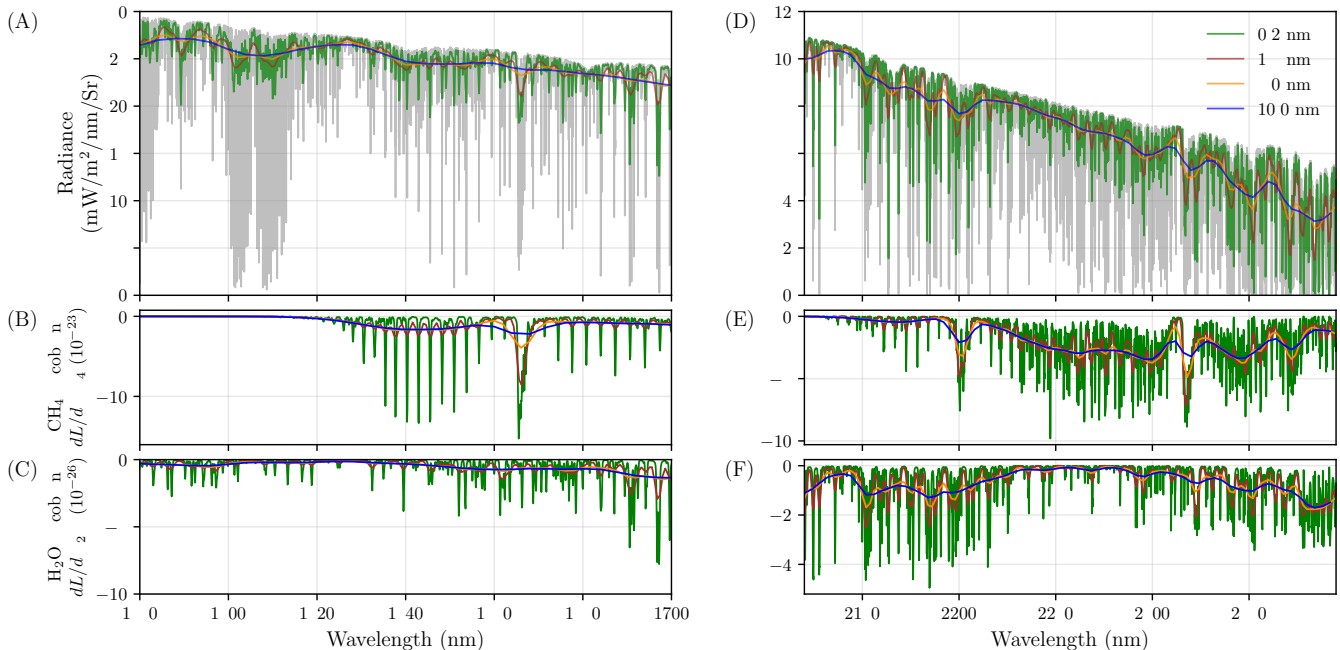

**Figure 5.** Simulated reflected solar radiance through a plume of $200 \, \text{kg} \, \text{hr}^{-1}$, over a concrete surface as observed by instruments of different FWHM in the two fitting windows of 1.6 μm (left) and 2.3 μm (right). The grey background is the originally calculated spectra at 0.01 nm.

## 3.2  Comparisons of two CH$_4$ fitting windows

We investigate the comparison between using two different CH$_4$ fitting windows near 1.6 and 2.3 μm. At 1.6 μm, incoming solar irradiance is higher, which could enhance the signal to noise ratio. However, at 2.3 μm the CH$_4$ absorption features are more numerous and prominent over a broader wavelength range, which should increase sensitivity for the CH$_4$ retrieval. Thus, there is a potential trade-off between the advantages and disadvantages of both retrieval windows. It is not immediately obvious which fitting window would result in a lower precision error. This could also depend on the spectral resolution, as

spectral fine-structure changes with increasing FWHM differently in both windows. Understanding this trade-off will help guide the development of future instruments.

The resolve which band is better suited for minimizing CH$_4$ retrieval errors also depends on typical surface albedos at 1.6 and 2.3 μm. We visualized the relative values of surface albedo at 1.6 and 2.3 μm across all surfaces in our database colored by their main surface categories in Figure 6. Most surfaces lie in the region where their albedos at 1.6 and 2.3 μm are relatively

equal (along the 1:1 line). Only a few surfaces have a much higher albedo at 2.3 μm compared to that at 1.6 μm, while there is a cluster of low-albedo surfaces for which the 1.6 μm albedo is about twice as high as at 2.3 μm. To compare the two fitting windows, we take two extreme surface examples: one with albedo of 0.87 at 1.6 μm and 0.14 at 2.3 μm, and another one with





albedo of 0.21 at 1.6 μm and 0.79 at 2.3 μm. We also consider a representative surface with an equal albedo of 0.51 at both 1.6 and 2.3 μm.

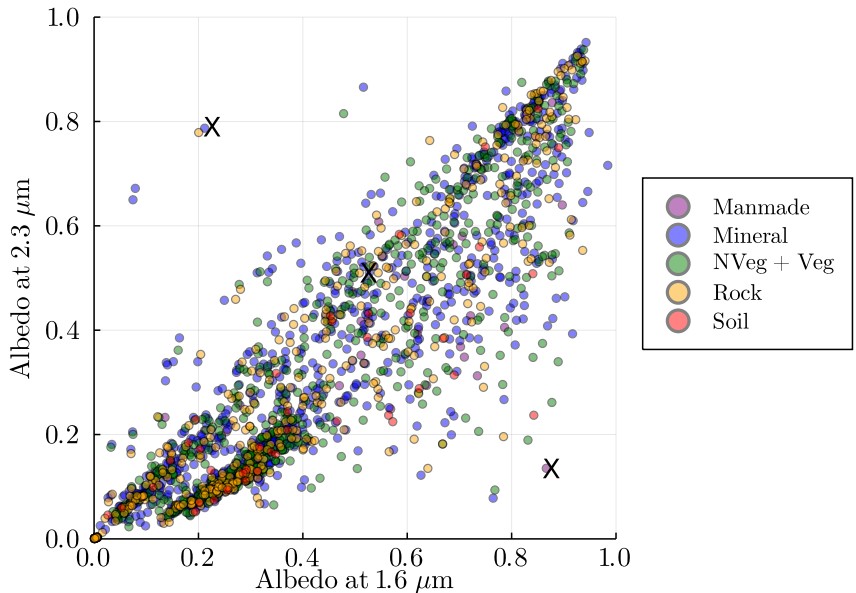

**Figure 6.** Scatter plot showing the relative values of reflectance at 1.6 vs 2.3 μm for different surface types. Each point is a distinct surface, and the color shows the type to which it belongs. Cross marks represent three example surfaces that we used in the comparison analyses.

In Figure 7, we compared the precision errors based on the two fitting windows over the three surfaces using a polynomial degree of 25, using fitting windows as displayed in Figure 5. In this simulation, only FWHM is varied and an oversampling of 2.5 is used (i.e. FWHM = 2.5*SSI). Other instrument parameters are set according to Table 1. The corresponding NESR of each detector pixel is then computed using the instrument noise model, enabling us to compute the changing noise levels with instrument resolution. We note that the number of detector pixels used varies considerably in this simulation, as they correspond to the window length (275 nm in the 2.3 μm band) divided by SSI. Thus, FHWM smaller than 0.65 in the 2.3 μm band would require more than 1000 detector pixels, which can be hard to achieve from a detector point-of-view, especially at fast readout rates, as required for high spatial resolution. This can preclude very fine spectral resolution if a large spectral bandwidth is required, which can be beneficial, especially if it allows additional species to be measured (such as $CO_2$). In addition, at coarser spectral resolution the full well capacity of the detector might be reached, which puts an upper limit of maximum SNR values per detector pixels (maximum SNR of 1000 for a full well capacity of $10^6$ electrons).

Here, we observe that the precision error is actually not monotonically improving with finer spectral resolution, as the SNR deteriorates in these cases. For the 2.3 μm band, the optimum is actually around 0.2 nm, which may appear surprising. For darker surfaces, the precision using a 1nm resolution can be equal to an instrument with 0.05nm resolution. Interestingly, the precision errors using the 1.6 μm band deteriorate much more with increasing FWHM than using the 2.3 μm band. The





reason for this is related to the methane band structure in both windows, as can be seen in the Jacobians (see Figure 5). In the 1.6 μm band, most of the fine-structure in the P and R branches are lost once the FWHM is coarser than the separation of the rotational fine-structure, leaving only the Q-branch spectrally distinct, even at lower spectral resolution. This could also be seen in SCIAMACHY, for which the FWHM in the 1.6 μm band was relatively low, namely about 1.3 nm (Frankenberg et al., 2005b). With the loss of spectral fine-structure, the precision deteriorates more rapidly as low frequency variations in

the Jacobians can be confused with surface albedo features, thus not directly constraining methane abundances. The situation in the 2.3 μm band is different as the absorption structures are very different, covering many more methane features that are irregularly spaced. Thus, some unique bulk methane absorption features persist even at spectral resolutions as coarse as 10 nm. In fact, this is the reason methane could be observed with imaging spectrometer at similar resolutions (Thorpe et al., 2014), enabling high spatial-resolution mapping from airborne instruments.

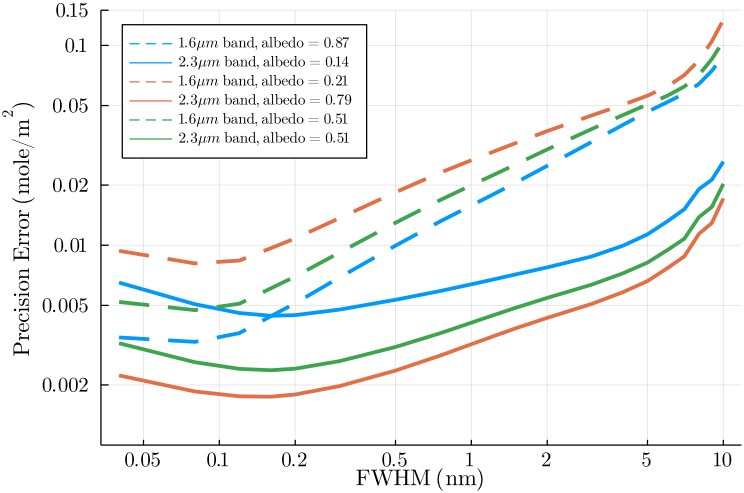

**Figure 7.** A plot showing precision errors from $CH_4$ retrieval in the 1.6 μm band (dotted) and the the 2.3 μm band (solid) over three distinct surfaces (colored), using instruments with different FWHM.

Overall, our retrievals in the 2.3 μm band consistently yield lower precision error compared to that from the 1.6 μm band. This difference is most highlighted for a surface with a much higher albedo at 2.3 μm compared to 1.6 μm. This is because decrease in solar irradiance at 2.3 μm is now compensate for by increased albedos, resulting in observed radiances to be relatively close to that at 1.6 μm, allowing for the effect of absorption depth and structure to be the sole driving force for a better performance. This effect is still seen for most typical surfaces of equal albedos at both 1.6 and 2.3 μm, where the

precision error from the 2.3 μm band is consistently lower than that from the 1.6 μm regions for instruments with any FWHM. The only scenario where the retrieval at 1.6 μm could perform better is under an extreme example where a surface has a much stronger albedo at 1.6 μm compared to at 2.3 μm and when FWHM is lower than 0.2 nm. These results indicate that the stronger and broader absorption of $CH_4$ in the 2.3 μm fitting window plays a more dominant role in the retrieval performance





compared to the stronger solar irradiance (and sometime higher albedo) in the 1.6 μm case. Since the latter condition for a
better performance in the 1.6 μm band is much more unlikely, we focus our following analysis on the 2.3 μm fitting window.

### 3.3  Spectral fit and error analysis with various instruments

Here, we evaluate the instrument performance of hypothetical yet realistic spectrometers covering the 2.3 μm range. If we
restrict ourselves to the number of spectral pixels in a fast detector, as used for AVIRIS-NG (480 spectral pixels), we can
achieve a FWHM of around 1.5 nm, minimizing the precision error under the hard constraint of a limited detector size. It
would also still allow for joint retrievals of $CO_2$ at 2 μm, as envisioned in Strandgren et al. (2020). This resolution allows us to
still resolve significantly more spectral fine-structure than current measurements at 5 or 10 nm resolution, which often exhibit
retrieval interferences with surface features, as seen in Figure 1. Our primary focus here is to quantify the impact of spectral
resolution on the ability to unequivocally separate surface spectral features from those in the atmosphere.

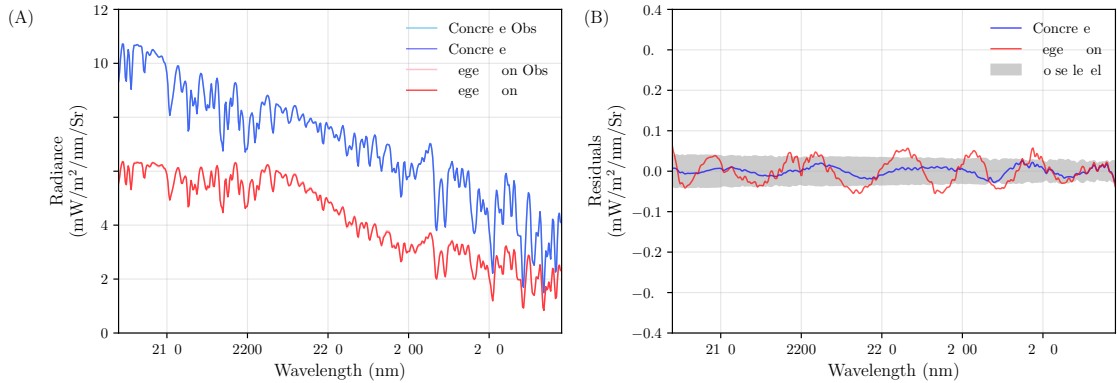

**Figure 8.** (A) The observed spectra through the exact same $CH_4$ source over two different surfaces, construction concrete and vegetation,
as observed by an instrument with FWHM of 1.5 nm and their best spectral fit. (B) Their associated spectral residuals and an expected 1-$\sigma$
noise level for the instrument measurement.

To demonstrate the impact of surface reflectance on observed spectra, Figure 8A shows two reflected spectra through the
exact same $CH_4$ concentration profile as observed by the same instrument with FWHM of 1.5 nm, but over construction
concrete compared to a vegetation surface. Evidently, not only do the observed spectra change in absolute value, but also their
spectral variations are different within the fitting window, more complex for vegetation than for concrete. This exemplifies an
important role that surface albedo plays in the retrieval, potentially interfering with the methane absorption lines. To further
validate this point, Figure 8B shows examples of residuals from best spectral fits for each of these two spectra using IMAP-
DOAS as described in section 2.4.2, with a polynomial degree of 11. The noise level indicates the theoretical 1-$\sigma$ noise level,
expected from the instrument. In panel A, these residuals are hardly visible as they are close to the noise level. Clearly, the fit
quality is different between the two surfaces as evidenced by a higher residual for reflected sunlight from the vegetation surface
compared to a sample construction concrete. However, differences are subtle and might not be detectable if noise level are high





or fewer detector pixels available. In practice, given an observed spectrum, our retrieval needs to differentiate the atmospheric

feature from the surface feature in order to obtain the best estimate of methane enhancement with minimal bias and precision
error. The performance of the retrieval will be influenced by the observing instrument as well as the representation of surface
reflectance in our forward model.

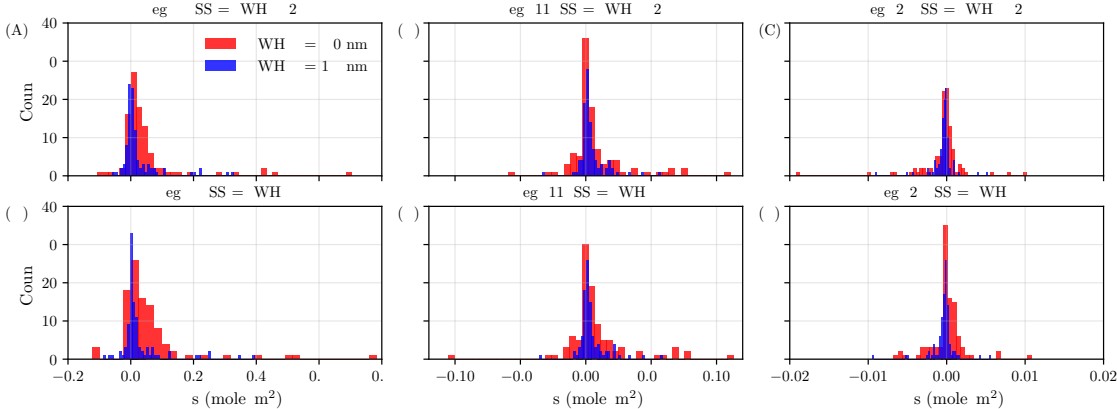

**Figure 9.** Plots showing a range of biases that occur over 100 randomly sample surfaces when different FWHM, SSI and polynomial degree
are used.

In this section, we performed the retrieval error analysis for different instrument parameters namely the FWHM and exposure
(integration) times. Since the surface albedo is another important factor in the retrieval performance as shown in Figure 8, we

also explored using different degrees in the polynomial terms in our forward model (see section 2.4.1), as a large number of
polynomial degrees is required to capture the effect of surface albedo spectral variability (e.g. a polynomial degree of 11 still
caused subtle yet systematic residuals for vegetation). For a given choice of instrument specification and polynomial degree,
the IMAP-DOAS algorithm is performed to predict the column methane concentration, and thereby derive error estimates. We
kept the true methane concentration fixed to a given vertical profile near the source emission with a flux rate of 200 kg hr$^{-1}$.

First, to illustrate potential bias that could arise from a variety of surfaces in real-world scenarios, we randomly selected 100
surfaces from our database and used each of them as a surface underlying the $CH_4$ column. By retrieving the observed radiance
using different instrument FWHM and different number of polynomial degrees, we analyzed the range of resulting biases for
each case. The bias distribution across the 100 surfaces are shown in Figure 9. When a polynomial degree of 5 is used in our
retrieval, the range of biases observed is (-0.2,0.8) mole m$^{-2}$, compared to just within (-0.15,0.15) and (-0.02,0.02) mole m$^{-2}$

with a polynomial degree of 11 and 25 respectively. This result suggests that some surfaces interfere strongly with the retrieval,
leading to very high biases when a low number of polynomial degrees is used, as this causes a forward model error. This
implies that the use of a higher polynomial degree has a significant consequence in minimizing the bias in our retrieval results,
especially when we have no prior information about surfaces in the vicinity of emission sources. A polynomial degree of
about 25 seems to capture most but not all surface effects. However, an instrument such as AVIRIS(-NG) only has 26 (52)





detector pixels covering the entire fitting window, thus not allowing us to use such high polynomial degrees as it would render the problem under-determined. This clearly illustrates the problem in separating surface and atmospheric feature at coarse spectral resolutions, as the problem becomes increasingly ill-posed with coarser spectral resolution. This is an integral part in obtaining reliable detection and quantification of local methane sources at a global scale, as is also shown later in section 3.4.1. In general, an instrument with smaller FWHM leads to smaller observed biases as expected by its ability to capture more high-

frequency $CH_4$ absorption features. Furthermore, with an SSI of FWHM/2.5, biases using higher polynomial degrees show a smaller range compared to the case with an SSI=FWHM, as surface features are harder to discern if atmospheric feature are not oversampled. In the following analyses, we therefore primarily show the results from the case of SSI=FWHM/2.5, unless otherwise stated. This also fulfills the Nyquist sampling requirement for typical atmospheric retrievals, which might sometime involve spectral shifts. The impact of spectral shifts is ignored here but would be another reason for both oversampling and

higher spectral resolution.

Next, we investigated how the precision error varies with instrument parameters such as FWHM and the exposure (integration) time. While FWHM governs shape of the methane Jacobians, the exposure time is an important factor determining potential spatial resolution and SNR. Figure 10 shows this relationship using a fixed exposure time of 20 ms with varying polynomial degrees on panel A and with fixed polynomial degree (25) and varying exposure time in panel B. Other parameters

are set according to Table 1, and the underlying surface is a construction concrete (albedo index 1 in Figure 3). If we vary the polynomial degree, the impact on precision is negligible for FWHM<0.5 nm, very small for FWHM<1.5 nm but diverging for FWHM>2 nm. The reason for this effect is that the polynomial degree determines which spectral variations can be purely attributed to methane and which might be caused by surface features, which eliminates its use to constrain methane. At fine spectral resolution, the methane fit is mainly driven by the atmospheric high frequency structure. Thus, the polynomial degree

plays no significant role. At coarser spectral resolution, most atmospheric features are blurred and can be partially confused with the surface, causing a divergence of the precision error with increasing FWHM. A 1.5 nm FWHM still allows us to sample sufficient atmospheric fine-structure to minimize the impact of surface interferences and will achieve sub-% precision error for a wide range of exposure times.

If we vary exposure times, the precision error decreases with the increase of the exposure time as the device can collect more

photons resulting in an overall stronger signal. There exists a range of FWHM values that minimize the precision error for each exposure time. Generally, the range lies between 0.1 - 0.3 nm. At low exposure times, readout noise becomes increasingly important and leads to a larger precision change when exposure times are varied. Thus, it is vital that an appropriate value of FWHM is chosen in order to achieve low precision error while we use a high degree of polynomial such as 25 in our retrieval to simultaneously reduce the bias from surface interferences. It is interesting to note that the FWHM with the best precision

moves towards higher FWHM with decreasing exposure times, being a consequence of the increasing role of readout noise at the detector.

For example, using FWHM of 1.5 nm and exposure time of 25 ms, a precision error of 0.007 mole m$^{-2}$ can be achieved, which is about 1% of the background total column amount of 0.7 mole m$^{-2}$. In low-earth orbit with a satellite speed of 7 km s$^{-1}$, 25 ms corresponds to a spatial resolution of 175 m. Spacecraft nodding would allow us to slow down the effective





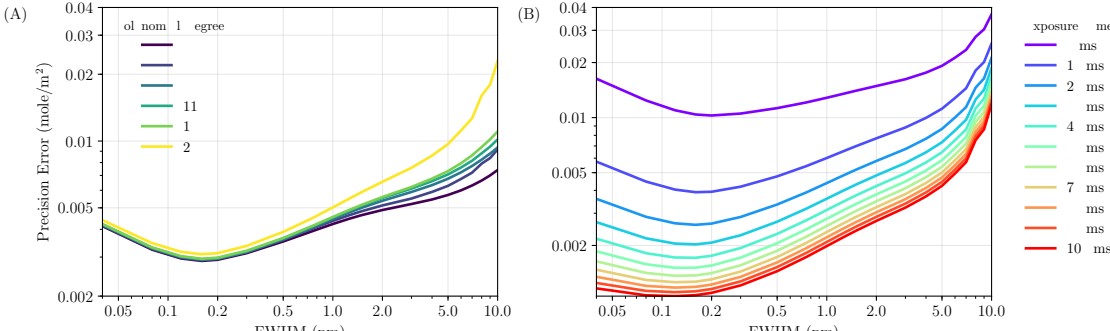

**Figure 10.** (A) Precision error as a function of instrument FWHM for different numbers of polynomial degrees used in the retrieval (with exposure time of 20 ms). (B) Precision error as a function of instrument FWHM for different exposure times (with polynomial degree of 25).

ground-speed by about a factor 10, rendering a 1% total column precision for $<20\,\mathrm{m}$ spatial resolution feasible from space using existing fast-readout SWIR detectors. This would be equivalent to measuring a pure methane layer of only 0.16 mm thickness.

### 3.4   2D retrieval over realistic surfaces

In section 3.1, 3.2, and 3.3, we have analyzed the impact of surface interferences and instrument specifications on the quality

of the $CH_4$ retrieval. The choice of parameters such as FWHM, exposure time and the number of polynomial degrees leads to significantly different precision errors and biases. These errors in each retrieval column ultimately affect the detection and quantification of $CH_4$ source in 2D scenes observed over various geographical areas across the globe. In this section, we illustrate how the retrieved $CH_4$ plume appears using different instruments and retrieval choices.

#### 3.4.1   Occurrence of false positives and false negatives

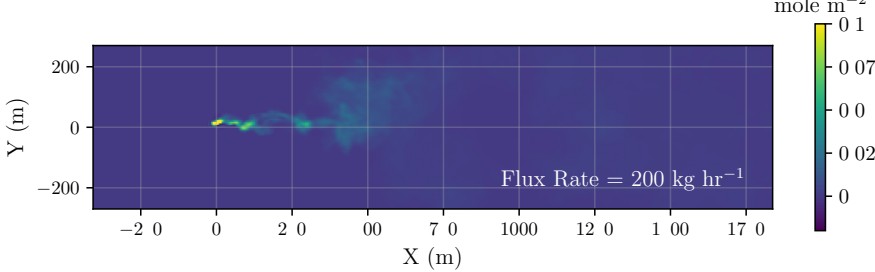

**Figure 11.** A nadir view of a simulated $CH_4$ plume from the LES with a prescribed flux rate of $200\ \mathrm{kg\ hr^{-1}}$. The background wind speed is $4\ \mathrm{m\ s^{-1}}$ along the x-axis.





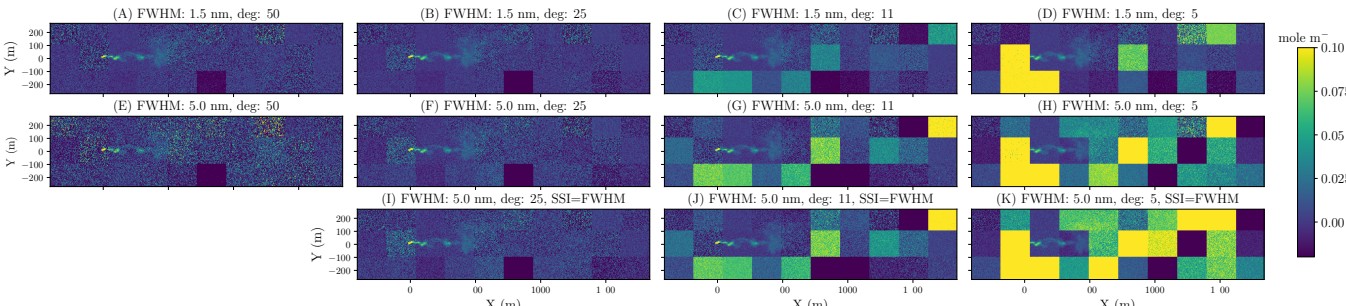

**Figure 12.** Retrieved $CH_4$ column over 30 different surfaces with varying instrument scenarios. The first row shows the results for an instrument with FWHM of 1.5 nm, while the second row shows the results for an instrument with FWHM of 5.0 nm. Both the first and second rows have SSI=FWHM/2.5. The third row shows the results for an instrument with FWHM of 5.0 nm and SSI=FWHM. The biases drop significantly as polynomial degrees increase.

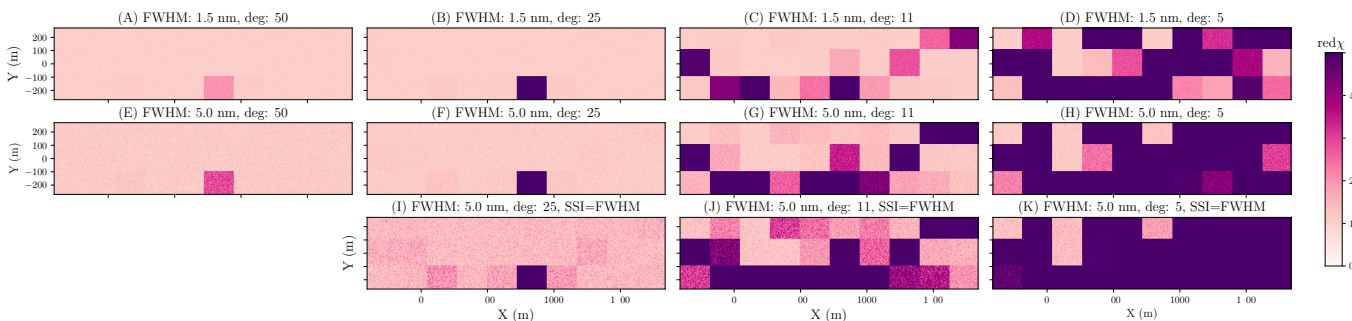

**Figure 13.** A plot showing the values of reduced $\chi^2$ from the retrieval of different instrument FWHM and polynomial degrees. As the polynomial degree becomes higher, the reduced $\chi^2$ decreases, implying a better spectral fit.

To explore the 2D pattern of a retrieved methane plume over a variety of realistic surfaces, we overlaid an LES simulated $CH_4$ plume on top of a checkerboard-styled land consisting of 30 surfaces. We then apply the instrument kernel and IMAP-DOAS algorithm to retrieve column $CH_4$ pixel by pixel across the 2D scene. A combination of FWHM of 1.5 and 5.0 nm, with an SSI of FWHM/2.5, and polynomial degrees of 5, 11, 25 and 50 are adopted. A case of SSI=FWHM is also used for a FWHM of 5.0 nm, being equivalent to the AVIRIS-NG instrument. The emission rate of the $CH_4$ source is equal to 200

kg hr$^{-1}$. The spatial distribution of the true $CH_4$ distribution is shown in Figure 11, showing local enhancements than can exceed 10% of the total background atmospheric column. The corresponding retrievals under different surfaces and instrument scenarios are shown in Figure 12. The deviation of predicted $CH_4$ from the true column value in each pixel is a combination of precision error and bias. In this result, the overall mean enhancement that emerges over each surface type in contrast to the true plume in Figure 11 could be interpreted as retrieval bias, while the presence of a speckle-like texture over each surface

can be viewed as the retrieval precision error driven by instrument noise for a given surface albedo (larger for dark surfaces).





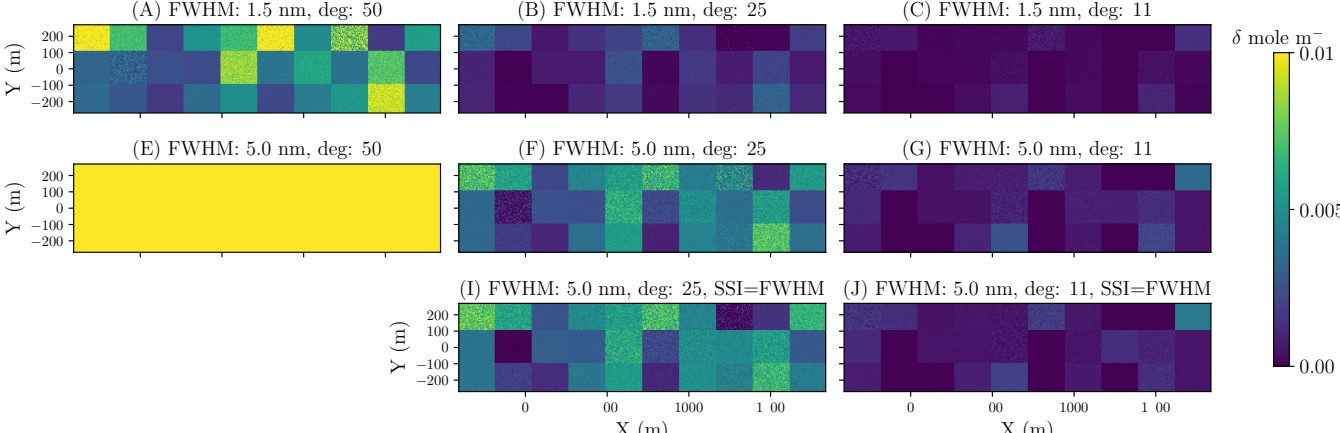

**Figure 14.** The increase in precision errors when the polynomial degrees of 50, 25, and 11 are used as compared to the polynomial degree of 5 case. The increase is computed for a given FWHM and SSI according to each row.

Meanwhile, the bias is related to systematic shift in the retrieved methane column enhancement from the true value due to surface interference in resolving the methane absorption features. In general, based on the visualization in these 2D plots, the more the retrieved enhancement scene resemble the true $CH_4$ concentration map, the better the performance of the instrument and retrieval is. Specifically, when a low polynomial degree of 5 was used in the retrieval, we observed significant retrieval

biases (both positive and negative) over various surfaces. Evidently, these biases can act to deceive the true location and enhancement of actual methane plume, especially if the surfaces have elongated shapes like a plume (not like a checkerboard here). As the number of polynomial degrees in the retrieval increases, the level of biases decreases over the scene enabling the actual methane plume enhancement to be better identified. This is also manifested in the reduced $\chi^2$ values which describe how well an observed spectrum was fitted (the smaller, the better). As shown in Figure 13, the value of reduced $\chi^2$ drops from 5 to

become increasingly closer to 1 when a polynomial degree is changed from 5 to 11, 25, and 50. We note that the 10 surfaces in the bottom row of this checkerboard-styled tile from left to right are the 10 surfaces with their albedo spectral variations shown earlier in Figure 3. The Pink Quartzite ($6^{th}$ on the list of these 10 surfaces) is an extreme case where we see an unusually strong variation near 2.2 μm, resulting in a persisting bias even at a polynomial degree of 25. Nevertheless, generally when a polynomial degree of 25 is used, most of the biases across surfaces seem to disappear but surprisingly complex surfaces such

as Quartzite can occur across various natural landscapes and human-made surfaces in cities.

At the same time, using higher polynomial degrees result in higher precision errors as can be seen from the speckle-like texture in the retrieved scene. To illustrate how precision deteriorates as we increase the polynomial degree for a given instrument FWHM, Figure 14 shows the rise in precision error when using polynomial degrees of 50, 25, and 11 relative to using the polynomial degree of 5. We can clearly see that precision error deteriorates with a higher polynomial degree, particularly at

coarser FWHM. This is consistent with what we observed earlier in section 3.3. Based on this analysis, we found that using an instrument with a FWHM of 1.5 nm would allow for higher polynomial degrees such as 25 to be utilized with a relatively small





increase in precision error. For an instrument with a FWHM of 5 nm, using a polynomial degree of 25 or 50 results in a larger precision error increase, underlining the potential problems that could occur when complex surface albedo features exist. The main physical reason for the deterioration of precision for low spectral resolution instruments is that a lower degree polynomial

in the fitting routine is equivalent to a hard a priori constraint that only spectrally smooth surfaces exist. The retrieval itself then attributes some of the broad-band variations in the methane Jacobian to be only attributable to changes in methane, not surface albedos, thus providing a tighter constraint on the methane abundances. For higher resolution instruments, most of the information content for methane is located within the fine-structure of the methane absorption lines and less on the broad-band variations, causing a much smaller increase in the precision errors if higher polynomial degrees are used.

These results indicate that a narrow FWHM (such as 1.5 nm) and a high number of polynomial degree (at least 25) are needed to reduce both precision errors and biases due to surface interference. This way we can obtain a higher-quality retrieved scene in order to effectively identify and quantify emission sources over the majority of the surfaces. If low polynomials are used, high biases are likely to occur over certain surface types. These can cause false positives or negatives in the observational systems, complicating the analysis of the locations and the emission rates of the $CH_4$ sources. This can cause a significant

problem for both human analysts as well as an AI models (Jongramrungruang et al. 2021 (in review)) that rely on the spatial distribution of observed enhancement to make predictions.

To further show an example over a real-world high emission area, we queried a realistic surface distribution over a well pad in the Durango area, Colorado from LANDSAT. At each location, this dataset provides surface albedos at the wavelengths of 0.48, 0.56, 0.65, 0.87, 1.61, and 2.20 μm. The RGB image of this particular location and its corresponding albedo near 2.2

μm are shown in Figure 15. Based on the albedos in the 7 bands available in this LANDSAT scene at each pixel, we found the best matching surface in our high-solution surface database from the ECOSTRESS spectral library. We used this surface to simulate a semi-realistic $CH_4$ scene with an emission emerging from the ground in the vicinity of a well pad. The retrieved $CH_4$ enhancement is shown in Figure 16. Once again, when an instrument with FWHM of 5.0 nm is used in conjunction with with a low polynomial degree of 5, high biases occur over most surfaces across the area. In particular, oil shale and sandstone

are the two surfaces that exhibit strong surface interference as observed in Figure 16(F). Evidently, the resulting bias occurs at a level that dwarfs the true plume enhancement rendering it impossible to identify the location of the emission sources, let alone the ability to obtain an accurate quantification of total emission in the area. However, by using a higher polynomial degree of 25, the biases across surfaces is greatly reduced, albeit with slightly higher noise. Nonetheless, this noise is reduced when FWHM decreases from 5.0 nm to 1.5 nm. We note that this reduction in precision error when we decrease the instrument

with a FWHM from 5.0 to 1.5 nm could be even more apparent over some other surfaces with lower albedos compared to the ones in this scene. Visually, we can already see that the actual $CH_4$ plume in terms of its location and strength can be much more easily identified and distinguished from the surface artefacts with the FWHM of 1.5 nm and polynomial degree 25. This finding demonstrates how achieving low bias and precision error in the observation and the retrieval process across diverse surfaces profoundly benefit the detection and quantification of true $CH_4$ sources. This analysis provides an insight on

how future instruments can be designed to enable an effective and accurate $CH_4$ source detection and quantification across





the globe. In the next section, we further illustrate retrieval performance when a local $CH_4$ plume of various emission rates is observed by an instrument with a higher spatial resolution such as 30 m.

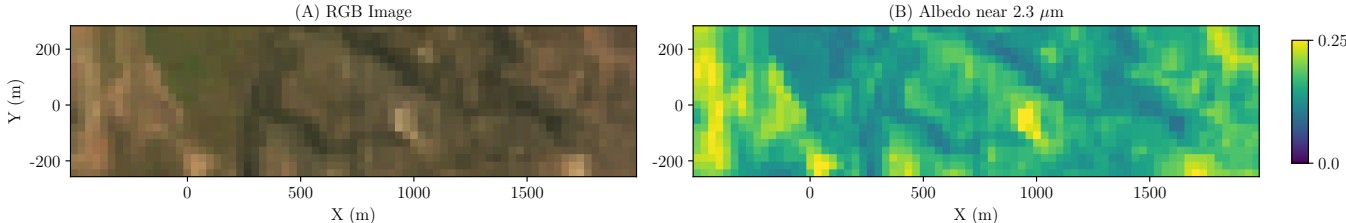

**Figure 15.** RGB and 2.2 μm albedo images of a realistic surface distribution in the Durango area, Colorado from LANDSAT.

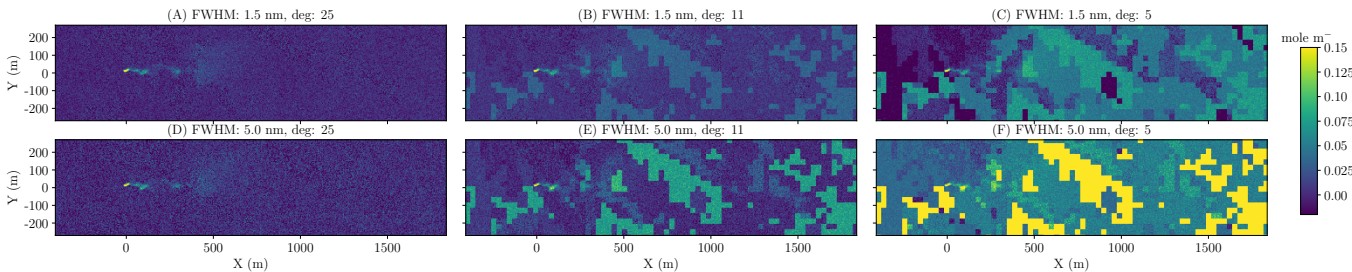

**Figure 16.** Retrieved $CH_4$ column over a well pad in the Durango area, Colorado area with varying instrument FWHM and polynomial degrees in the retrieval. The realistic surface distribution is based on LANDSAT data. The use of polynomial degree of 25 and FWHM of 1.5 nm can reduce biases and precision error over this scene.

### 3.4.2 Effect of spatial resolutions and flux rates

In the previous section, we have shown a 2D retrieved scene assuming that an observing instrument has a spatial resolution of
5 m. In this section, we repeated the 2D scene retrieval analysis with a spatial resolution of 30 m by averaging the reflected sunlight through a $CH_4$ plume simulated at 5 m spatial resolution into 30 m spatial resolution prior to applying an instrument operator. We present this analysis with this design consideration for the 30-m spatial resolution and the exposure time of 40 ms to evaluate the potential of future spectrometers on-board satellites in the coming years. The 2D scenes retrieved at 30m spatial resolution by instruments of different FWHM and polynomial degrees are illustrated in Figure 17. The choice of FWHM of 1.5
nm and a polynomial degree of 25 remains very effective in removing surface biases across the scene and the location of $CH_4$ plume can be distinguished. It is important to note, however, that the retrieved $CH_4$ column concentration near the source pixels becomes more diluted as the spatial resolution decreases. This is because the local $CH_4$ plume distribution at an emission rate such as $200 \, \mathrm{kg \, hr^{-1}}$ varies greatly on scales of just a few meters. Having demonstrated that an instrument with a FWHM of 1.5 nm and a polynomial degree of 25 can significantly reduce precision error and biases due to surface interference, we use





this setup to investigate how the 2D retrieved scenes look like for sources of different emission rates to understand the lower limit of CH$_4$ emission rates that can potentially still be detected.

The retrieved scenes for CH$_4$ emission rates from 50 to 2000 kg h$^{-1}$ are shown in Figure 18, and the corresponding scenes showing the ratio of retrieved methane concentration and precision error in each pixel are given in Figure 19. The ratio, $n$, of pixel enhancement to precision error represents a $n-\sigma$ probability event that this pixel enhancement would have randomly 495   happened, purely due to noise, in the absence of a true CH$_4$. A ratio value above 4 would imply that there is only a probability of lower than 1 in 15000 that this would happen by chance due to random noise. Thus the ratio of 4 can be a simple and useful metric to imply where an actual CH$_4$ enhancement pixels are. Based on this metric, we found that CH$_4$ source detection can still be possible for plumes with emission rates as low as approximately 50-100 kg h$^{-1}$.

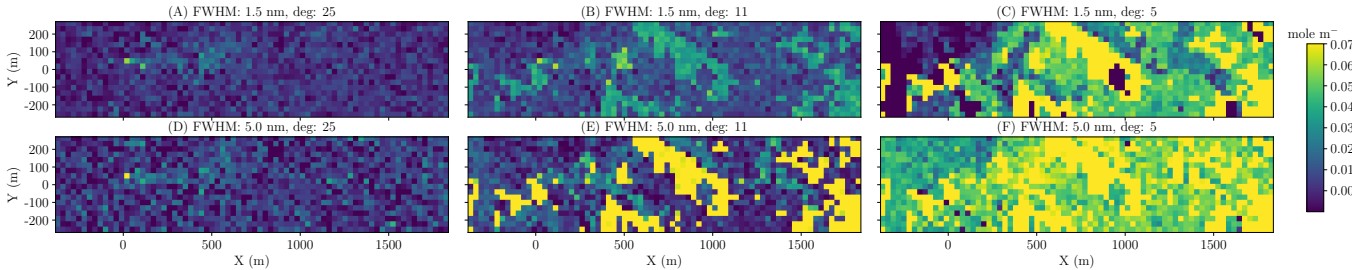

**Figure 17.** A plot similar to Figure 16 but at 30-m resolution. Retrieved CH$_4$ column over a well pad in the Durango area, Colorado area with varying instrument FWHM and polynomial degrees in the retrieval. The realistic surface distribution is based on LANDSAT data. The use of polynomial degree of 25 and FWHM of 1.5 nm can reduce biases and precision errors across the scene.

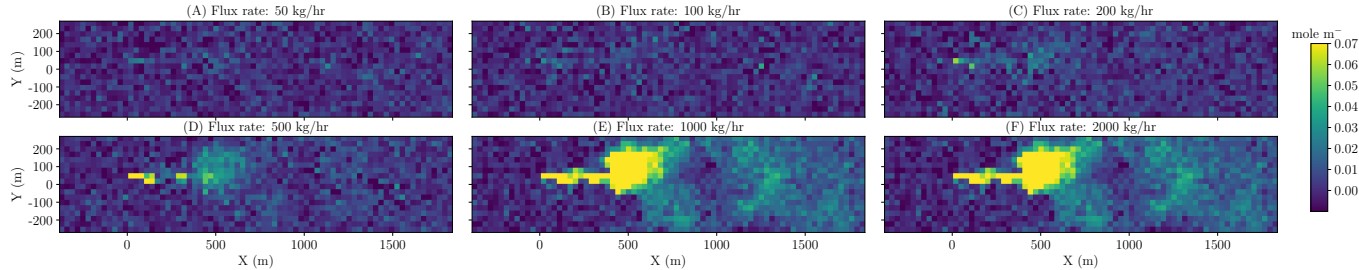

**Figure 18.** Retrieved CH$_4$ column over a well pad in the Durango area, Colorado area for plumes of various emission rates. The spatial resolution is 30 m, the instrument FWHM is 1.5 nm, and the polynomial degrees in the retrieval is 1.5. The realistic surface distribution is based on LANDSAT data. Column enhancement in the vicinity of CH$_4$ plume are increasingly visible as the source emission rate becomes larger.





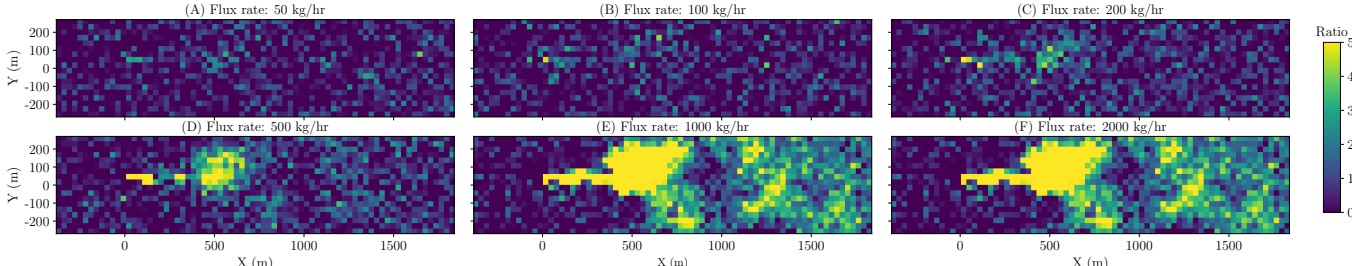

**Figure 19.** Ratio of retrieved $CH_4$ excess column divided by the posterior precision error, over a simulated scene in the Durango area, Colorado area for plumes of various emission rates. The spatial resolution is 30 m, the instrument FWHM is 1.5 nm, and the polynomial degrees in the retrieval is 25. The realistic surface distribution is based on LANDSAT data. Ratios higher than 4 implies a probability of lower than 1 in 15000 that the pixel enhancement happens by random noise.

## 4 Conclusions

We built an end-to-end modeling framework that can simulate radiances from reflected sunlight through methane plumes over a variety of surfaces. In this study, we simulated a realistic 3D $CH_4$ concentration field from a point source using an LES, and varied the underlying surfaces where the emission occurs using a comprehensive surface albedo database from the ECOSTRESS spectral library consisting of over 2000 surface types. The observed radiances and their noise-equivalent spectral radiance for various instrument configurations are modelled directly as a function of incoming radiance and instrument parameters such as the FWHM of the lineshape function and integration time, without having to prescribe the SNR a priori. Based on the modelled radiance, we applied the IMAP-DOAS algorithm to retrieve methane column enhancements.

We compared the tradeoff between the two most frequently used fitting windows for $CH_4$ in the 1.6 and 2.3 μm ranges. Our analysis has shown that despite a higher solar radiance near 1.6 μm, the stronger absorption feature of $CH_4$ near 2.3 μm leads to a consistently lower precision error for the 2.3 μm fitting band. The rare occasion of a 1.6 μm fitting window outperforming the 2.3 μm retrieval band happens only when both FWHM is lower than 0.2 nm and simultaneously a surface in consideration has a much higher albedo near 1.6 compared to 2.3 μm. For the purpose of building an instrument to detect methane emission accurately at sufficiently fine spatial resolutions across most global surfaces, we believed that the 2.3 μm band can perform better and should be prioritized in most scenarios. We primarily considered the fitting window in the 2120 - 2395 nm range to study the impact of instrument parameters and retrieval choices on the retrieval bias and precision error.

To highlight the impact of surface interferences, the number of polynomial degrees is varied in the IMAP-DOAS retrieval experiments. This framework allows us to derive the corresponding precision error and bias when different sets of instrument parameters and the number of polynomial degrees are used in the retrieval of $CH_4$ column concentration. Our analysis shows that the number of polynomial degrees used to represent surface spectral variations in the retrieval algorithm has a significant impact on the bias of the retrieved methane columns causing a positive bias as large as 0.8 mole m$^{-2}$ for retrieval with polynomial degree of 5 compared to 0.2 and only 0.02 mole m$^{-2}$ for degrees of 11 and 25, respectively across the majority of





surfaces. Using a higher polynomial degree, however, is found to simultaneously increase precision error for methane retrieval as this relaxes the constraints on possible methane absorption contribution in spectral variations of the observed radiance. This is particularly evident at FWHM of greater than $2\,\mathrm{nm}$. Thus, using an instrument with a lower FWHM such as $1.5\,\mathrm{nm}$ will allow for a high number of polynomial terms to be used while inducing a smaller deterioration in precision error. For example,

we found that an instrument with FWHM of 1.5 nm and the exposure time of 20 ms can achieve a precision error of less than $0.007\,\mathrm{mole\,m^{-2}}$ (or less than 1.0% of the total column in the atmosphere) over a typical construction concrete surface with an albedo of 0.35, even when a polynomial degree of 25 is used.

    Having low bias in the retrieval is integral to removing correlated surface features in the retrieval enhancement map. These surface features from retrieval errors likely appear as false positives and subsequently cause significant impacts on the detection

and quantification of true $CH_4$ sources. To demonstrate the significance of surface interferences, we used a realistic surface distribution over a well pad from the Durango area, Colorado as a background surface with an LES methane plume of 200 $\mathrm{kg\,hr^{-1}}$ to create a synthetic emission in a real world environment. Our results illustrated that when an instrument of high FWHM (such as 5 nm) is used with a low polynomial degree (such as 5), a large retrieval bias appears broadly across the 2D scene. These interferences occur severely over surfaces such as oil shale and sandstone, resulting in difficulties to clearly

distinguish a true plume from areas of systematic biases. Nevertheless, by using a lower FWHM value such as 1.5 nm and the polynomial degree of at least 25, we have illustrated the ability to obtain low retrieval bias across the entire scene and to effectively differentiate the source location from the background. We also repeated the 2D retrieval analysis for 30-m spatial resolution by averaging the radiance per unit area from $5\,\mathrm{m^2}$ to $30\,\mathrm{m^2}$, and adjusted the exposure time to $40\,\mathrm{ms}$ which could be achievable for future satellites. Again, our results have shown that using FWHM of 1.5 nm and a polynomial degree of 25

plays a crucial role in resolving surface features and removing false positives, ultimately enabling the ability to distinguish the true emission location. In the absence of bias, the ratio of retrieved column enhancement and the retrieval precision error in our retrieved 2D scenes indicates that it might be possible to detect a $CH_4$ emission source with a flux rate as low as $50\text{-}100\,\mathrm{kg\,h^{-1}}$.

    This study highlights the effect of changing instrument FWHM, exposure times and the polynomial degree on minimizing retrieval errors. The FWHM and exposure time are intrinsic to how a spectrometer is designed, as opposed to describing a

spatial resolution which depends on external factors such as the viewing geometry and the speed of a remote-sensing platform. Further studies will be required to translate how these variables are implemented into an observing system that can achieve specific spatial resolutions of interest. Additional considerations such as the device saturation constraint will also influence the ultimate achievable exposure time for a newly designed instrument. Our end-to-end simulator that includes an instrument model and retrieval can be generalized to study the performance of future instruments with specific engineering requirements.

The findings in this study can inform future satellite instrument designs and the retrieval algorithm in order to have robust observations capable of separating real plumes from surface interference. Reducing both bias and precision error can have a profound benefit both for manual analysis by humans and for an automated model plume detections such as an Artificial Neural Network approach. This will enable the analytic chain to have a higher accuracy and level of confidence detecting and quantifying more subtle methane sources from observed scenes across large geographical areas.



The modelling framework in this work could also be generalized to improve the detection and quantification of $CO_2$, with minor modifications on a different fitting window and a different magnitude of flux rate in the emission simulations.

*Author contributions.*  SJ and CF conceptualized and designed the research objectives. SJ performed the analysis, and wrote the paper. GM ran the LES model, and provided output. CF provided guidance on the overall design and support on scientific approaches and experimental setups. AT and ZZ provided feedback and suggestion on figures and text. All co-authors contributed to the writing of this paper.

*Competing interests.*  The authors declare that they have no conflict of interest.

*Acknowledgements.*  This work is part of SJ's NASA Earth and Space Science Fellowship (NESSF, grant no. 80NSSC18K1350). We acknowledge the Resnick Sustainability Institute at Caltech for their kind support with computing resources. We deeply thank Rupesh Jeyaram for his kind support with a radiative transfer open-source software (https://github.com/RadiativeTransfer), and his help with making our computations run faster.



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
