# Peer review of "Remote sensing of methane plumes: instrument tradeoff analysis for detecting and quantifying local sources at global scale"

_Atmospheric Measurement Techniques, 2021_

## Author Comment (AC1)

**Reviewer 1**

We thank the reviewer for the constructive comments and appreciate the review.

Please find a point by point response below:

A. General Comments

Atmospheric methane is the second important greenhouse gas, but their emission estimate from different source sectors has large uncertainties. Imaging spectrometers using large number of pixels is a powerful tool to detect CH4 plumes. Their selection of spectral range and resolution and integration time impact the performance directly.

Existing works mainly study trade-off between instrument noise and spectral resolution using $CH_4$ absorption spectra only, which results in too optimistic results. In real measurements, surface albedo estimation is one of the major error sources. The simulation tool and analytical methods in this paper are realistic and present consistent results.

Objectives of the $CH_4$ measurements with a new instrument seem to be both detection of unknown emission source and estimation of emission quantitatively. The former is well written, but the latter is not clear. It will help readers' understanding to list at least the possible error sources and discuss how to reduce uncertainty briefly.

In this work, we focused on demonstrating how a better choice of instruments can significantly reduce the retrieval error of methane concentration in each pixel. This directly leads to an improvement in both the detection and quantification of methane emissions, which is not a part of the current study. For the quantification aspect, typical methods rely on the total enhancement of methane plume around the source pixels (referred to as IME in Varon et al. 2018, Jongaramrungruan et al. 2019). These studies show that if high retrieval errors occur due to surface interference near the source, it could significantly add to the uncertainties in the total enhancement, and therefore emission rate estimates (such as a falsely high flux rate for an actual small source). However, when retrieval error is minimized, we could remove this source of error from the flux inversion step, resulting in a more reliable flux estimate down the line. The most important factor is to randomize and de-correlate error sources so that aggregate emissions estimates are not biased.

I recommend publication after minor revision.

B. Specific Comments

(1) Page 2, line 32, "methane emission"

It not clear. Methane emission of what?

"For instance, just the question whether or not the leak rate in the natural gas extraction system is 1 or 2% is equivalent to a 100% uncertainty in methane emissions."

By this, we mean that estimation of methane emission rates from the natural gas extraction system can be highly uncertain due to the fact that the overall leak rates from these systems are not precisely known. We adjusted the text in the manuscript to be "For instance, just the question whether or not the leak rate in the natural gas extraction system is 1 or 2% is equivalent to a 100% uncertainty in methane emissions from these leaks" for more clarity.

(2) Page 5, Incoming solar irradiance,

Just a comment. Recently published paper "The TSIS-1 Hybrid Solar Reference Spectrum" 10.1029/2020GL091709, discussed uncertainty in the continuum at 1.6 and 2.3 micron regions and includes Toon's line spectra.

We appreciate the reviewer pointing to this paper. We will look into the possibility of using the new solar reference spectrum product in our future related work. For the current synthetic sensitivity study, the choice of the solar model will have a minor impact.

(3) Page 13, Figure 6 caption

Brief description of the selected surface area will help readers' understanding. For example, "our database of different surface albedos from the ECOSTRESS spectral library".

We have added this additional description to the caption of Figure 6 accordingly.

(4) Page 18, 3.4.1. Occurrence of false positive and false negative

Larger degrees of polynomial provide better fit. However, too many retrieval parameters also produce larger errors. Authors mention that the optimized degree depends on the spectral resolution of the instrument. This paper described in detail. Once the design is fixed or when readers already used their existing imaging spectrometers, it will be very helpful if there are index or criteria to determine the optimized degree of polynomial.

As the reviewer mentioned, in a situation where the instrument specification is fixed, an optimized degree of polynomial can be found based on the spectral observations and its fit. In this work, we highlighted the benefit of using higher-resolution FWHM in the instrument to allow for higher degree of polynomial to be used. In the future, methods such as backward elimination could potentially provide a real-time determination of the number of polynomial degrees needed, for a given instrument over a certain surface type.

C. Technical Corrections

(1) Page 3, line 84

The sentence "Hence the origination of this study" looks incomplete.

We modified the sentence to be "This motivates the origination of this study."

---

## Author Comment (AC2)

**Reviewer 2**

This paper presents a synthetic study on the retrieval of methane plumes from satellites with high spatial resolution. This is a quickly developping area and a number of satellite (and aircraft) instrument have emerged that have succesfully demonstrated methane retrievals on a scale of tens of meters. Such observations will be important to detect and mitigate methane emisisons from localised emission sources. However, it is critical to put such methane satellite retrievals on a solid footing. This study addresses the question how well surface features and methane absorption can be seperated which is a key issue for instrument with lower spectral resolution. This is relevant for ongoing work with existing satellites but more importantly it provides guidance for the development of of future mission. The manuscripit is suitable for Atmos. Meas. Tech. and I recommend publishing it after addressing my comments below.

We thank the reviewer for the constructive comments and appreciate the thoughtful review.

Please find a point by point response below:

Figures: many figures in the manuscript are corrupted. This is probably simply an issue of the pdf conversation.

We apologize that this issue occurred. We did not hear of this problem from other reviewers, but will make sure that the figures are clearly shown in the publication, using only Vector graphics.

Instrument assumptions: The study provides a realistic model for the instrument and the measuerement noise calculation. The model makes uses of a number of instrument parameters given in Table 1. Can you please provide a justifcation of these assumptions. How does this compare to currently available systems and existing detectors. Is the assumption valid that the same parameters can be used for the two spectral range (1.6 and 2.3 micron): will detector quantuum efficiency, grating efficiency, spectral transmissivity/reflectivity of optical components not change between both ranges? Also, at 2.3 micron, I would assume that thermal emission of the optical bench will be a contributor to noise. Can you give some example values for dark current to support your assumption that this can be ignored. Finally, can you please clarify if noise has been added to the simulated spectra (Figure 8B suggest otherwise).

Most detector characteristics are in line with state-of-the-art detectors such as the Teledyne Chroma series (http://www.teledyne-si.com/products/Documents/CHROMA%20Brochure%20-%20rev%201%20v5%20-%20OSR.pdf). With cryo-cooling, dark current is very low and its impact on noise can mostly be neglected (also the thermal emissions from the optical bench). QE of

the detector can indeed be wavelength dependent but the impact is small. Some publicly available can be found in https://www.teledyne-e2v.com/content/uploads/2018/10/ICSO_2018_Teledyne_IR_Sensors_PJerram_JBeletic.pdf (also consistent with assumptions made in the Strandgren paper).

Noise has indeed been added to the spectra.

Surface features and polynomial degree: A key outcome of the study is the need for a very high degree of a polynomial to sufficiently accurately describe surface features. However, the use of a polynomial of degree 50 makes me uneasy. Can you show with a direct polynomial fit to the underlying surface albedo data of the ECOSTRESS spectral library before using it in your forward model and without any spline interpolation that such a polynomial degree is needed? I would also expect that a high polynomial degree will lead to an increased number of non-converging retrievals when not carefully choosing their a priori value and a priori covariance; can you please elaborate on your choice. As you show in the paper, a high polynomial degree will increase the retrieval uncertainty for methane. At the same time the correlation with methane will increase so that you risk that the methane absorption will be taken out by the polynomial. Did you have a look at the correlation coefficients ?

We applied a polynomial degree of 50 as an illustration of how it could change bias and precision error. Over most surfaces, a polynomial degree of 25 seems adequate to significantly reduce the retrieval error (both bias and precision error combined).

The polynomial fits for an example surface (construction concrete) are shown in the following figures. The surface albedo in this demonstration is without any spline interpolation. Legendre polynomials were used and the range of wavelength is 1400-2500 nm. Even when the x-axis is adjusted to be in between -1 to +1 to make the fit easier, we clearly observed that high degrees such as 25 or 50 were needed to capture the surface albedo variations.

[Figure]

[Figure]

[Figure]

Fit with deg 50

In our experiments, we set a prior value in our polynomial degree coefficients as 0.5, -0.1 for the first two degrees and 0.0 for the remaining degrees, and we use a loose prior covariance (1e20) to have no significant impact on the retrieved total columns or posterior errors. Based on this setup, the non-converging retrievals did not arise in our experiments.

Minor comments and typos: I have included them directly in the supplementary pdf.

We make edits throughout the text in response to the minor comments accordingly.

Fast detector: Enabling exposure times <50ms of a large focal plane array (to allow high spatial resolution, unlike say TROPOMI, which can integrate for 1s, which makes the readout easier as well)

And the unit of readout noise of 100 electrons.